# Are Respiratory Responses to Cold Air Exercise Different in Females Compared to Males? Implications for Exercise in Cold Air Environments

**DOI:** 10.3390/ijerph17186662

**Published:** 2020-09-13

**Authors:** Michael D. Kennedy, Elisabeth Lenz, Martin Niedermeier, Martin Faulhaber

**Affiliations:** 1Faculty of Kinesiology, Sport and Recreation, University of Alberta, Edmonton, AB T6G 2H9, Canada; kennedy@ualberta.ca; 2Department of Sport Science, University of Innsbruck, 6020 Innsbruck, Austria; Elisabeth.Lenz@student.uibk.ac.at (E.L.); martin.faulhaber@uibk.ac.at (M.F.)

**Keywords:** exercise induced bronchoconstriction, winter sport athletes, wheeze, post exertion cough, airway hyperresponsiveness, spirometry, athlete health, airway provocation

## Abstract

Research has shown that cold air exercise causes significant respiratory dysfunction, especially in female athletes. However, how female and male athletes respond to cold air exercise is not known. Thus, we aimed to compare acute respiratory responses (function, recovery and symptoms) in males and females after high-intensity cold air exercise. Eighteen (nine female) athletes completed two environmental chamber running trials at 0 °C and −20 °C (humidity 34 ± 5%) on different days in a randomized starting order. Spirometry was performed pre, 3, 6, 10, 15 and 20 min post. Respiratory symptoms were measured posttrial and heart rate and rating of perceived exertion were assessed during each trial. No significant differences in delta change (pre to post) were found at either temperature between sexes for FEV_1_, FVC, FEF50% and FEF25–75%. At −20 °C, FEV_1_ decreased similarly in both sexes (males: 7.5%, females: 6.3%) but not at 0 °C, *p* = 0.003. Postexertion respiratory function recovery and reported symptoms were not different between sexes at either temperature. These results indicate no sex-based differences in acute respiratory responses (function, recovery and symptoms) to cold air exercise. However, intense exercise at −20 °C is challenging to the respiratory system in both sexes and may lead to altered respiratory responses compared to mild winter conditions like 0 °C.

## 1. Introduction

It is well understood that in some individuals, exercise-related respiratory complaints [1,2] occur and are associated with transient airway diameter constriction during exercise [3]. Collectively, this phenomenon is described as Exercise-Induced Bronchonstriction (EIB), defined as a “transient narrowing of the lower airway after exercise” [4] found in elite [5] and habitually active populations [6] and often exacerbated by cold air compared to warm air exercise in the same individual. Field-based cold air studies have been employed [7,8,9] to understand EIB; however, such studies are limited in elucidating the relationship between air temperature and EIB onset and/or severity due to the lack of standardized cold air conditions. For that reason, standardized air temperature studies have been employed to better control the inhaled air temperature and humidity conditions.

In the few well-controlled environmental chamber studies to have been undertaken, respiratory function was shown to be suppressed [10] and respiratory symptoms increased in healthy adults during cold air exercise [11,12,13]. These initial studies [11,12] used only one cold air temperature condition (i.e., either −18 and −23 °C), while others using a controlled cold air environment found that EIB was greater at temperatures colder than −15 °C compared to 0 °C [13] in female participants. The latter study also found that female participants engaging in cold weather “High-Ventilation (HV)” sports had a greater EIB response, a common finding in HV athletes of both sexes compared to non-HV athletes [13]. This is because inhaled cold air is also dry, and the drying of the airway is the most potent mechanism leading to EIB [14,15], inducing chronic respiratory airway hyper-responsiveness [16] over years of cold exposure in HV athletes [14]. Respiratory symptoms are also commonplace after intense cold air exercise where “post-exertion cough,” [17] wheeze, chest tightness and excessive mucus are most frequently observed [7,18,19]. Furthermore, the time course of spirometry measures returning to baseline (i.e., pre-provocation values) has been found to be prolonged with progressively colder air temperatures in females [13], a finding which has not been evaluated in males and females in the same study. 

Thus, the accumulated evidence to date does not provide clear evidence on how females differ in their acute response to cold air exercise compared to males. This is surprising, given that the female lungs are more mechanically constrained, inducing more shear stress in the airway at the same exercise intensity compared to males [20]. Some limited descriptive studies have shown that adult female athletes have significantly greater self-reported respiratory symptoms in a variety of HV sports compared to male athletes [21,22]; however, sex-based respiratory symptom differences have not been researched extensively. Recent evidence also indicates that prepubescent, male Nordic athletes (less than 13 years of age) have a greater self-reported wheeze than females, but this finding was related to greater training hours in males compared to females [23]. A recent field-based study did find a more severe EIB response in females than males at −15 °C [15]; however, the majority of known sex-based differences in respiratory function are based on warm weather or normal laboratory conditions [24,25]. Thus, there is enough preliminary evidence to hypothesize that sex-based differences in EIB in cold environments may exist. Furthermore, this study also addresses the current paucity of sex-based comparisons in exercise science [26], and hopes to improve sex-specific recommendations in sport science practice [27]. Therefore, we aimed to assess acute respiratory responses (function, recovery and symptoms) after intense exercise in healthy female and male participants at both mild and cold ambient air temperatures (0 °C and −20 °C) in a controlled setting. We hypothesized that female participants would show greater reductions in respiratory function, have prolonged recovery and report more respiratory symptoms than male participants. 

## 2. Materials and Methods 

### 2.1. Participants and Experimental Design

Eighteen (9 female) healthy, habitually active university students who were used to vigorous exercise bouts were recruited. Participants were required to have a maximal oxygen uptake (VO_2max_), i.e., >50 ml/kg/min for males and >45 ml/kg/min for females. Study participants were prescreened during their first visit using an adapted PAR-Q (Physical Activity Readiness Questionnaire) [28]. The questionnaire also included questions about heightened symptoms of EIB and cough during cold-weather exercise and a previous history of adverse respiratory symptoms due to high-intensity exercise (regardless of the temperature at which the adverse respiratory symptoms occurred). Exclusion criteria included pre-existing acute or chronic diseases, heightened symptoms of EIB and cough during low or moderate intensity exercise in cold weather, a previous history of adverse respiratory symptoms due to high intensity exercise, regular smoking, i.e., >5 cigarettes/day and pregnancy or lactation. All participants had the opportunity to ask questions at any given time over the course of the study and provided written informed consent prior to any testing. No a priori power analysis was performed for the present study; however, a sensitivity analysis was conducted for the outcomes of acute respiratory function using G*Power 3.1 (University of Düsseldorf, Düsseldorf, Germany) [29]. Assuming alpha = 0.05 (two-tailed), power = 0.80 and an independent samples *t*-test, an effect size of d > 1.40 is revealed significant with the present sample size of 18 participants. The study received approval from the Board for Ethical Questions in Science, University of Innsbruck (certificate number: 12/2018). Subsequently, all eighteen subjects completed the baseline examination protocol as follows: A baseline spirometry assessment was completed prior to a graded maximal treadmill test. During the graded maximal treadmill test to fatigue, expired gas analysis (Oxycon Pro, Care Fusion, Germany) and heart rate (RS800, Polar, Kempele, Finland) were measured continuously in ambient laboratory conditions (23 °C; Relative Humidity range: 40–70 %). The rating of perceived exertion (RPE) (Borg-scale 6–20) was taken every minute and immediately after the participant stopped [30]. The graded maximal treadmill test protocol was adopted from Kennedy et al. [13], and consisted of 3-min stages, starting at 6 km/h and increasing by 1 km/h at a standard grade of 5% until exertion. The last completed stage represented the target speed for the 8 min of high-intensity exercise trials in the environmental chamber. 

After the baseline examination day was completed, participants were randomly assigned to their first cold-air exposure. There was a minimum of 48 h between baseline examination and the first cold air trial. The second cold air trial was spaced at least 24 h after the first, and all participants completed the study in the same 3-week period in the months of March and April. Participants were allowed to continue their normal exercise and activity patterns during that time. However, participants were asked to refrain from performing any high-intensity exercise for 24 h prior to each test. Other pretrial guidelines included abstaining from the consumption of heavy meals and alcohol on days when a trial was scheduled.

### 2.2. Cold Air Trials

All participants were informed one day prior to their cold-air trial of the temperature in which they were going to perform. Before entering the environmental chamber, pretrial spirometry was completed and participants were equipped with a heart rate strap (RS800, Polar, Kempele, Finland) to measure heart rate during the trials. The severe exercise protocol in the environmental chamber included a standardized warm up of 5 min’ easy walking at the target air temperature to reduce any cold pressor effect that might occur [31]. A 10-min individual warm-up at 1 % grade at a freely chosen speed was next. The selected speed documented in the first cold air exercise trial was used in the second trial. Heart rate was measured every 2 min and RPE was recorded at the end of the warm-up. This warm-up protocol is based on a standardized exercise pattern used in a previously published study [13]. Subsequently, a transition minute started in which the 1 % incline was gradually increased to 5% and the participants’ individual severe exercise target speed was set. The individual target speed was defined as the speed of the last completed work stage during the graded maximal treadmill test to fatigue. For the next 8 min, participants were asked to maintain that overall velocity at 5% grade; this was designed to achieve approximately 90% maximum heart rate. The overall work load was chosen based on previous research that showed heavy ventilation as a result of 8 min of running with a 5% grade and a speed that induced VO_2peak_ [13,32]. During these 8 min, RPE was taken at minutes 1, 3, 5 and 7 and the heart rate was measured continuously. Additionally, participants were asked for a global perception of the workload immediately after the cold air trial using a single item scale ranging from Rest (0) to Maximal (10). Participants spent a total of 24 min in the cold-air chamber completing each cold air trial. At the end of the 8 min of all-out running, participants had the opportunity to walk on the treadmill for another 1 min before exiting the environmental chamber and performing the post-trial spirometry at minutes 3, 6, 10, 15 and 20, based on previous methods [12].

Each cold-air exposure took place in a digitally controlled, custom environmental chamber (Siemens AG, Munich, Austria) for temperature and humidity. A customized treadmill (h/p/ cosmos sports & medical GmbH, Nussdorf, Germany) on which the participants performed their exercise bouts was set up inside the environmental chamber, which has an ambient operating range of 35 °C to −25 °C (see Figure 1). For each trial, two investigators were present: One was in the control room operating the treadmill (adjusting grade and speed) and ensuring the consistency of the temperature and humidity level. Additionally, the person in the control room assessed the overall completed distance during the 8-min heavy ventilation stage for each cold-air trial, since the velocity had to be reduced in some cases due to participants’ fatigue. This reduction in speed was set at 0.5 km/h and documented manually. To ensure that every participant finished the 8-min stage, the speed was reduced by another 0.5 km/h on top of the initial reduction if necessary. The second person was in the environmental chamber together with the participant, recording heart rate and RPE manually. 

Participants were allowed to wear temperature appropriate clothing designed for exercise outdoors including a toque and gloves. Furthermore, participants could add and/or discard clothing during the warm-up period in order to reduce the chance of thermoregulation influencing their performance during the 8-min stage of heavy ventilation exercise. However, at no point were participants permitted to cover their face or mouth in any manner (scarf, buff, hand) or tuck their chin into their collar cuff.

### 2.3. Respiratory Function Testing and Questionnaires 

Spirometry was completed on a portable electronic spirometry device (SP1; Schiller, Linz, Austria) that was validated against standardized calibration protocols using a 3 L syringe in our research department. All spirometry tests were conducted by trained personnel according to the American Thoracic Society (ATS) guidelines in an upright sitting position to ensure consistency [33]. Post-trial spirometry took place at an ambient temperature of between 20 °C and 25 °C for each trial. During the short waiting periods between the post-trial spirometry assessments at 3, 6, 10, 15 and 20 min, participants were allowed to walk around in order to imitate a cool-down period after exercising. The pre- and post-trial spirometry measures recorded included forced vital capacity (FVC), forced expiratory volume in the first second (FEV_1_), forced expiratory flow at 50% (FEF50%) and forced expiratory flow at 25–75% (FEF25–75%). 

To understand respiratory and other symptoms associated with cold air exercise, participants were asked to fill out a questionnaire. The questionnaire was administered with the goal of capturing the frequency of the four most common respiratory symptoms associated with EIB in the 24 h after cold air trial: cough, wheeze, chest tightness/trouble breathing (dyspnea) and excessive mucus secretion [18,34], as well as a global rating of effort for that trial. 

### 2.4. Calculation of Variables and Statistical Analysis

Statistical analysis was performed using SPSS Statistics version 24 (IBM, New York, NY, USA). In the first step, possible sex differences in participant characteristics at baseline were tested using *t*-tests for independent samples for normally distributed data (Shapiro-Wilk test) and Mann-Whitney U-test for non-normally distributed data. 

The main analysis was to assess possible sex differences in acute respiratory responses (function, recovery and symptoms). All analyses on sex differences were conducted separately for the two temperatures, i.e., 0 °C and −20 °C. Two outcomes were calculated for acute respiratory function: absolute maximum change in FEV_1_ and FVC, both of which were calculated using the minimum observed values after the cold air trial (i.e., the time point after exercise which presented the maximum delta)—the values prior to the trial. Relative maximum change was then determined by dividing absolute delta by the pretrial value × 100, as per previous recommendations [35], to determine prevalence and severity of EIB (where at least a 10 % drop in FEV_1_ was considered EIB [36] and the magnitude of the EIB was determined as mild, moderate or severe, as per the guidelines for EIB severity [10]). Subsequently, the relative changes in FEV_1_ and FVC were analyzed according to sex differences with an independent samples *t*-test. Cohen’s d was reported as an effect size for acute respiratory function parameters [37], with negative values indicating larger changes in females compared to males. Several outcomes were tested for respiratory recovery post-trial. We calculated the absolute changes in spirometry measures (FVC, FEV_1_, FEF50% and FEF25–75%) from all post-trial time points—the pretrial value. Relative changes were then calculated by dividing absolute delta by the pretrial value x 100. The relative changes were analyzed using a 2 × 5 mixed analysis of variance with the within-subject factor time (reduction from pre- to 3-, 6-, 10-, 15- and 20 min post-trial) and the between-subject factor sex (female and male). Partial η^2^ was reported as an effect size for respiratory recovery post-trial. The Greenhouse–Geisser adjustment was used to correct for violations of sphericity (Mauchly test). For the analysis of sex differences in the number of symptoms, a Mann-Whitney U-test was used and Cohen’s d was reported as an effect size. Sex differences in the frequency of specific symptoms (cough, wheeze, chest tightness and mucus) were analyzed by Χ^2^ tests. The odds ratio was reported as an effect size for frequency of symptoms with values > 1 indicating a higher frequency of symptoms in females compared to males. Additionally, three exercise intensity measures were analyzed for sex differences at the different temperatures. The mean of both maximal heart rate (HR max) and reported perceived effort (RPE) was calculated. Subsequently, HR max, RPE and global perception of the workload were compared between sexes using *t*-test for independent samples, since all data showed normal distribution. Cohen’s d was reported as an effect size for exercise intensity measures. 

In a secondary analysis, we analyzed differences in the variables for acute respiratory function and number of symptoms between the temperatures, 0 °C and −20 °C, in the total sample. A Wilcoxon-test was used in this comparison, since the difference values did not show normal distribution. Cohen’s d was reported as an effect size. 

Descriptive analysis is described as means ± standard deviations and/or absolute/relative frequencies. For all analyses, the alpha level was set at *p* < 0.05 (two-tailed), meaning that any value below this number was considered significant.

## 3. Results

Participant characteristics for males and females are shown in Table 1. Males were significantly taller, heavier, had greater VO_2max_ and maximal speed, as well as larger spirometry values. The physical activity and exercise histories were similar among all participants, and all participants were regularly engaging in intensive exercise bouts/intervals as part of their university degree requirements or club-based sport activities in the Tirol region of Austria. The participants were all living in the same geographic area and experienced similar environmental exposure to outdoor ambient conditions. All participants resting baseline spirometries were within normal range for their age, height and sex for FEV_1_ and FVC, but 4 female participants’ baseline FEV_1_/FVC ratios were less than < 0.75, which is predictive of exercise-induced bronchoconstriction [15]. 

### 3.1. Spirometry Measures 

The outcomes for acute respiratory function are shown in Figure 2. Relative changes expressed as percentage from pretrial values did not show significant sex differences for FVC at 0 °C, *p* = 0.115, d = −0.83, or at −20 °C, *p* = 0.328, d = 0.14, FEV_1_ at 0 °C, *p* = 0.690, d = −0.12, and at −20 °C, *p* = 0.273, d = 0.17. Regarding EIB identification (expressed as percent relative change = delta/pre-trial × 100) at 0 °C, one female participant and one male participant had a decrease in FEV_1_ between ≥10 to <25%, which indicates mild EIB [10]). All other participants at 0 °C either had a decrease in FEV_1_ between 0 and 10 % (1 female and 4 male) or increased FEV_1_ postexercise (7 females and 4 males). At −20 °C, 3 female participants had a mild EIB response (range = 12 to 23 % decrease), as did 3 male participants (range = 11 to 16 % decrease). Three female and 6 male participants had a FEV_1_ decrease of between 0 and 10 %, which indicates a “normal EIB response”. At −20 °C, 4 female and 5 male participants had a greater reduction in FEV_1_ compared to 0 °C.

No significant temperature difference was found for FVC between −20 °C (−6.0 ± 4.1%) compared to 0 °C (−2.3 ± 8.5%), *p* = 0.327, d = −0.55. The relative change in FEV_1_ was significantly lower at −20 °C (−7.1 ± 1.7%) compared to 0 °C (0.7 ± 7.8%) when males and females were combined, *p* = 0.003, d = −1.38.

At 0 °C, the variables of respiratory recovery post-trial (relative pre-/post-trial changes in FVC, FEV_1_, FEF50 and FEF25–75%) revealed no significant effect of sex, *p* > 0.094, part. η^2^ < 0.17, indicating similar respiratory recovery in females and males (Figure 3). Relative change in FVC at 0 °C showed the largest nonsignificant sex difference. Furthermore, no significant sex by time interaction was found, *p* > 0.205, part. η^2^ < 0.10. Significant time effects were found for the change in FVC, *p* = 0.048, part. η^2^ = 0.19, and the change in FEV50%, *p* = 0.029, part. η^2^ = 0.15. The change in FVC showed an increase over time, and the change in FEV50% dropped from 3 to 6 min, at which point it stayed relatively constant. 

At −20 °C, the variables of respiratory recovery post-trial revealed no significant effect of sex, *p* > 0.604, part. η^2^ < 0.02, indicating similar respiratory recovery in females and males (Figure 4). Furthermore, no significant sex by time interaction was found, *p* > 0.110, part. η^2^ < 0.13. Significant time effects were found for the change in FVC, *p* < 0.001, part. η^2^ = 0.13, and the change in FEV_1_, *p* = 0.009, part. η^2^ = 0.26. Changes in both FVC and in FEV_1_ showed an increase over time. 

### 3.2. Respiratory Symptoms Post-trial

No significant differences between female and male participants in the number of respiratory symptoms at 0 °C and −20 °C were found post-trial (0 °C: *p* = 0.796; −20 °C: *p* = 0.796) (see Table 2). The overall reported number of symptoms (males and females combined) after the exercise trials was significantly greater at −20 °C (2.4 ± 0.8) compared to 0 °C (1.2 ± 0.9; *p* = 0.002). The most common respiratory symptom post 0 °C exercise trial was mucus production for females (78%) and cough for males (44%). At −20 °C, 78% of female and 89% of male participants reported cough, while 78% of both female and male participants reported chest tightness/trouble breathing and 67% of female and 78% of male participants reported mucus (see Table 2). However, no significant sex difference was found in the incidence rate of the specific symptoms (cough, wheeze, chest tightness/trouble breathing and mucus) at 0 °C or −20 °C (Table 2). 

### 3.3. Exercise Intensity Measures 

When comparing female and male maximal heart rate results, no significant sex difference was found at 0 °C (*p* = 0.762, d = −0.15) or −20 °C (*p* = 0.494, d = 0.33). There was no significant change in maximal heart rate in female participants at −20 °C (187.5 ± 8.0 bpm) compared to 0 °C (187.0 ± 9.0; *p* = 0.834, d = 0.06), but maximal heart rate in males was lower at −20 °C (185.0 ± 6.0) compared to 0 °C (188.0 ± 7.5; *p* = 0.094, d = −0.44). The rating of perceived exertion (RPE) showed no significant sex differences at 0 °C (*p* = 0.245, d = 0.52) or −20 °C (*p* = 0.538, d = 0.52). However, females reported significantly greater global workout values at 0 °C (8.8 ± 0.8) compared to male participants (7.1 ± 1.6; *p* = 0.014, d = 1.34). At −20 °C, there was no significant difference between females (8.6 ± 1.0) and males (7.8 ± 1.9; *p* = 0.318, d = 0.53). The target speed of the treadmill during the 8 min of all-out running varied between 10 km/h and 16 km/h among all subjects. 

## 4. Discussion

This is the first well-controlled environmental chamber study that compares respiratory function, symptoms and perceptual responses to high-intensity cold air exercise at 0 °C and −20 °C in females and males. Due to previous research indicating that females are more susceptible to airway hyperresponsiveness (females 60% vs. males at 22%) and respiratory symptoms during exertion (19% vs. 12%) [21], it was hypothesized that intense exercise in the cold would suppress female respiratory function and increase symptoms more than in males, especially under colder conditions. We also aimed to determine whether the perceptual and physiological response to high-intensity exercise in females might explain any potential respiratory differences between genders. Despite our hypothesis, sex-based differences in acute respiratory responses (function, recovery and symptoms) were uncommon both at 0 °C and at −20 °C (see discussion below); however, both sexes did have significantly more constriction at −20 °C compared to 0 °C, which confirms the idea that with decreasing air temperature, greater airway constriction occurs. This validates previous research, which found that decreasing air temperature increases constriction in both mixed sex cohorts [12,39] or females [13]. However, it is important to note that exercise (i.e., cold air exercise) is an indirect airway provocation method [36,40] compared to direct methods, such as methacholine challenge, where sex-based differences in airway constriction have been found [21]. Extrapolating from previous studies, which also included indirect provocation tests (lab-based or cold air running test), they found no discernable sex-based differences, although they did not make a direct sex comparison [15]. Thus, our findings, in combination with others, indicate that the method of provocation may influence the diagnosis of EIB, and future studies should aim to understand sex-based differences to both direct and indirect provocation tests. 

Exploring our spirometry measures in more detail, it must first be noted that a pairwise comparison of any spirometry difference measured as pre- to 3, 6, 10, 15 or 20 min post-trial at 0 °C or −20 °C did not reveal any differences between sexes. Overall, this would support the conclusion that healthy male and female adults who engage in cold air exercise have similar postexercise respiratory function. Traditionally, EIB research has focused on absolute reductions (the maximum delta from pre- to post-trial) to understand the “maximum influence of the provocation”; this maximum delta can be expressed in absolute or percent change (max delta divided by the pretrial value) [9,41]. Although nonsignificant, the mean difference between sexes in relative FVC at 0 °C was relatively large when compared to that at −20 °C. A possible reason for this might be found in the significantly greater perceived global workout rating in females at 0 °C compared to males; a difference that was not evident at −20 °C. It is possible that the intensity was slightly higher in females at 0 °C, leading to a nonsignificant reduction in FVC in females compared to males. However, mean differences in FEV_1_ were negligible, and both FEF50% and FEF25–75% increased post-trial at 0 °C in both females and males, so these results are inconclusive regarding the question of whether an overall sex difference is apparent at 0 °C. 

Further, our findings at 0 °C align with those of previous research which found that a high intensity warm up induces bronchodilation at warm indoor temperatures (20 °C) [42,43] or outdoor mild temperatures of around −2 °C [44]. Our exercise protocol was similar in intensity and duration as that of other research which observed a refractory period after high intensity warm up, especially in EIB-negative individuals at a similar intensity and temperature [44]. This finding is important from an environmental human health standpoint, because it illustrates that exercise at around 0 °C may not cause the same degree of EIB as exercise below −15 °C, and in some instances, might have a positive effect on airway function (improved spirometry measures). Our results suggest that this response to exercise at 0 °C is similar in both males and females; however, future studies are needed to confirm this. At −20 °C, all relative changes in spirometry measures were decreased in both males and females, and the magnitude of the decrease was not greater in females, suggesting that females and males may have a similar responses to severe cold air exercise. However, future research with larger sample sizes is needed to detect more subtle sex-specific effects. We can also confirm from these results that severe cold air exercise, i.e., at <−15 °C, poses a significant challenge to the airway, leading to significant cold air-induced airway constriction, as shown numerous times before [7,8,22,39,45,46,47,48]. 

Exploring the baseline characteristics of our participants, some (4 females) were at greater risk of EIB based on a FEV_1_/FVC ratio of less than 0.75 [4,49]. Certainly, these participants all had reduced FEV_1_ post-trial at −20 °C but not at 0 °C, which furthers the argument that the severity of the cold air provokes greater constriction in those with underlying obstruction. From an environmental perspective, these findings highlight the fact that those with underlying obstruction measured as a low FEV_1_/FVC ratio should be wary of intensive cold air exercise. However, we cannot definitively conclude that females are more vulnerable to cold air exercise, because other factors such as height, bodyweight, total lung volume and cumulative exposure of high ventilation exercise in cold air can also affect the FEV_1_/FVC ratio. Thus, our results must be interpreted with caution, because it may be coincidental that it was all females with underlying obstruction (based on FEV_1_/FVC ratio) and the largest decreases in FEV_1_ at −20 °C. From an EIB identification standpoint, we would also characterize our results as not being sex dependent, because the same number of males had mild EIB as did females at both 0 °C and −20 °C. What is clear is that severe cold air illuminates diagnoses of EIB because 6 (3 female, 3 male) cases were found at −20 °C compared to 2 at 0 °C. Overall, the prevalence of EIB found at −20 °C in our study (33%) was similar to that of other research with similar participants exercising at cold air temperatures [15,39]. Given the fact that our participants were apparently healthy with no history of reported asthmatic or shortness of breath events during or after exercise (including those with a low FEV_1_/FVC ratio), our results confirm that in habitually active females and males, cold air-induced EIB likely occurs in 30 to 40% of individuals at −20 °C. 

Generally speaking, an individual’s symptoms associated with intense cold air exercise varies largely and is substantially influenced by a number of factors such as athletic discipline [50], overall training hours per week [51] or the geographic location (dry inland winter environments appear to lead to greater symptoms) [52]. In our study, all participants grew up in the same inland geographic region, and all were exposed to the same mountainous environment during the winter months preceding the study. This, potentially, would have increased the prevalence of respiratory symptoms, because cold dry air and mountainous environments have recently been cited as the most potent for the development of symptoms following exercise in athletes [53]. Relative to young athletic individuals in Sweden [23], our results show higher prevalence, but compared to athletes in Quebec, Canada, we observed the same prevalence of symptoms after exercise [50]. We also found the same pattern of respiratory symptoms as other research following a cold air race [18,51] or intense exercise [13], where cough was the most prevalent symptom compared to other symptoms (wheeze, chest tightness/trouble breathing and excessive mucus). From a sex differences standpoint, we found no significant sex difference in the number of respiratory symptoms at 0 °C and −20 °C post-trial (0 °C: *p* = 0.796; −20 °C: *p* = 0.796). Similarly, no significant difference was found in the incidence rate of the specific symptoms (cough, wheeze, chest tightness/trouble breathing and excessive mucus) between female and male athletes. Within sex, the total number of symptoms also increased proportionately from 0 °C to −20 °C, which aligns with previous research in females [13], although others have found no increase in symptoms with cold air running [39]. We partly predicated our hypothesis on previous research, which found self-reported respiratory symptoms were greater in females than males [21]; however, our results do not support any postexercise differences in the prevalence or type of symptoms. Contrasting how previous research findings differ from our results, we propose that our methodological approach might explain the discrepancies. Specifically, we looked at acute responses to cold air exercise, whereas others looked at the longer-term influence of cold air exercise exposure. In this context, our research indicates that total exposure to cold air exercise might affect females more than males, but in the short-term (i.e., within a single acute bout), females and males have a similar response. Future research should seek to confirm this conjecture. 

Regarding the time course of recovery post-trial from 3 to 20 min in female and male participants, neither a significant main effect of sex nor a significant sex by time interaction was found in spirometry measures at 0 °C and at −20 °C. Generally, the greatest reduction in respiratory function occurred at 6 min post-trial, except for FVC values, where the greatest reduction occurred at 3 min post-trial. Overall, these findings are supported by previous studies that have shown that the greatest reduction in respiratory function occurs within 5 min after cold-air exercise, with a return to baseline values at approximately 30 min [32,54]. However, the present study also indicates that there was a trend in the male participants towards complete recovery compared to female participants in the −20 °C condition (see Figure 2A,B). This highlights a potential sex-based difference related to respiratory function recovery that should be examined further. Reflecting on the lack of complete recovery, we acknowledge that the ambient laboratory conditions (approximately 20 °C) in which the post-trial measures were taken might have played a role. As McFadden et al. has illustrated [46], the inhalation of ambient laboratory air following cold air exercise can in fact increase the rate of water-loss from the airway (termed “burden of rewarming”), prolonging airway narrowing [55] and potentially increasing the time of recovery. Thus, the time of recovery might have been influenced by the indoor ambient air conditions in our study, and more importantly, the length of recovery might be greater in females than males. It would be important in the future to examine the post-trial ambient condition effect on respiratory function recovery, because this might provide some practical evidence on how soon an individual should return indoors after a cold weather exercise bout.

From a physiological and perceptual effort standpoint, our research contributes to our understanding of how females and males respond to the same intense exercise under both moderate and cold air conditions. With regard to the maximal heart rate, female and male subjects were similar, which is in line with the findings of other studies examining high-intensity exercise and respiratory function [56]. They achieved maximal heart rate during the 8 min of all out running in both trials (mean maximal heart rate values at 0 °C: females: 95%, males: 94%; −20 °C: females: 96%, males: 93% of HRmax), with no significant sex difference. Given that the heart rates we observed were between 92–97% of the maximum values, this is qualified as “severe intensity with an associated blood lactate concentration ranging from 6 to 10 mmol/L [57]”. This insinuates that all participants likely experienced hyperpnea at both temperatures, since they were above the respiratory compensation threshold that generally lies between 85 and 90% of the maximum heart rate in well-trained individuals [58]. Furthermore, we did not find any differences in acute RPE, i.e., both sexes rated the exercise bout as “very hard to extremely hard” [30]. Thus, overall, we have no reason to conclude that the perceptual responses to cold air exercise are sex-dependent, and thus recommendations for cold air exercise should be the same for females and males. 

From a limitations standpoint, although we recruited healthy participants from the same geographic region, the types of sports and recreational pastimes which the individuals engaged in were not recorded in great detail. This lack of understanding may have influenced the results, given that sport background may influence response to an airway provocation test [59]. We also did not collect spirometry data after the maximal exercise test as an indicator for severity of EIB in those conditions, nor did we complete an extensive atopy questionnaire such as the AQUA [60]. If we had completed an atopy evaluation, we could have understood if atopy played a role in the EIB response [61]. Moreover, the standardized severe exercise protocol in the environmental chamber was performed on a treadmill; therefore, these results are most pertinent to running related activities. Additionally, these participants were free-living individuals and would have had some variations in their physical activity patterns over the study period. Although we suggested that participants should not engage in any high intensity exercise in the 24 h before a test, it is possible that high levels of accumulated fatigue affected their exercise intensity. Finally, our sample size was similar to previous studies investing cold air exercise in an environmental chamber, yet a larger sample size may have improved our ability to understand the nature of sex-based responses to cold air exercise. 

## 5. Conclusions

Given that many nations in the northern hemisphere and nations with mountainous regions both have prolonged winter seasons with severe cold temperatures, these results are applicable from an environmental health perspective. To elucidate, we feel that these results can be generalized to both females and males who live in winter or mountainous climates and are habitually physically active outdoors during the cold weather months. Our key finding would be that the respiratory function, symptoms and perceptual responses are similar between sexes. This implies that females, who we hypothesized to be more vulnerable to cold air exercise, are not, and can engage in cold air exercise with similar responses as males. This is not to say that there was an absence of deleterious respiratory symptoms associated with severe cold air exercise (our −20 °C condition); it just means that we would advise both females and males to make healthy choices regarding their outdoor exercise activities, especially when the air temperature is −20 °C or less. Further, we would conclude that other factors which are known to be predictive of significant responses to cold air exercise should be considered, such as baseline FEV_1_/FVC ratio, where a ratio less than 0.75 was associated with EIB, and in this case, these participants were all female. Thus, if we were to propose any population that was more vulnerable, it would be individuals with a FEV_1_/FVC ratio of less than 0.75, and in all likelihood, most with this low ratio will be female. This is likely due to the stature effect where small stature females have greater shear stress overall in their lungs, leading to a heightened state of sensitivity to airway provocation [25], especially to cold air provocation [15]. In this case, small stature individuals who are otherwise healthy might consider applying a heat and moisture exchanger, which was shown to be effective in reducing the prevalence of EIB-associated dysfunction and respiratory symptoms such as cough [62]. 

## Figures and Tables

**Figure 1 ijerph-17-06662-f001:**
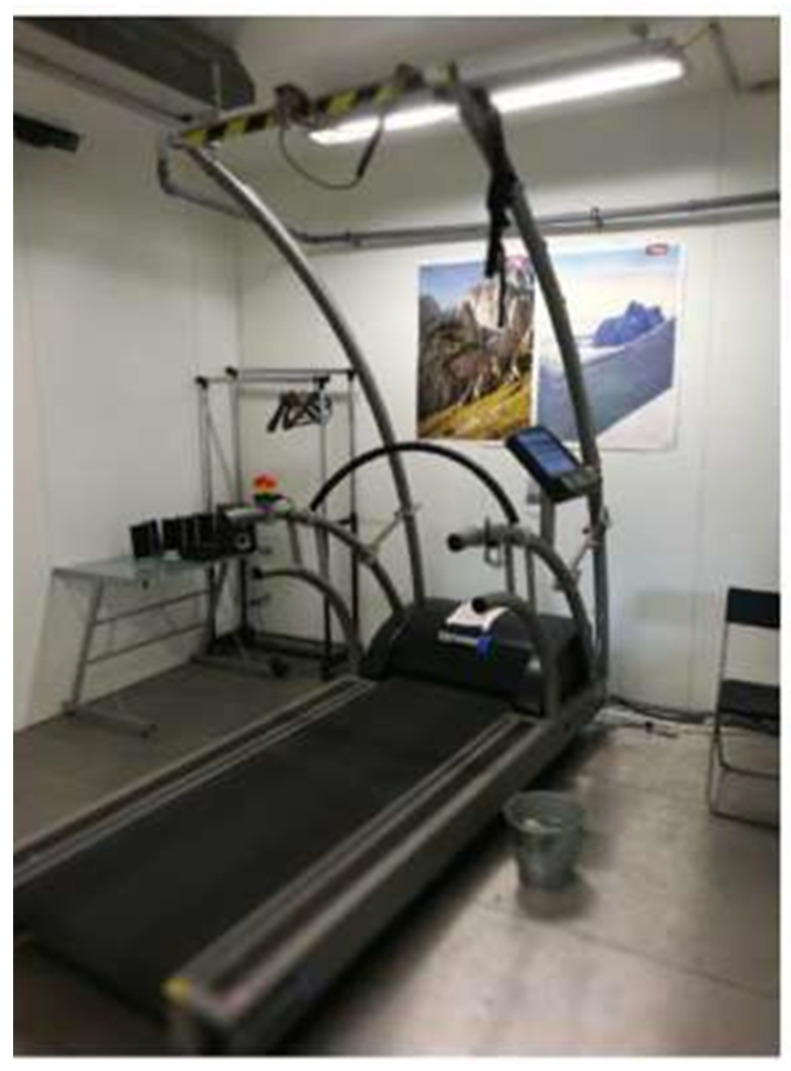
Interior view of the environmental chamber at the Department of Sport Science at the University of Innsbruck, Austria. Treadmill: h/p cosmos sports & medical GmbH, Nußdorf, Germany.

**Figure 2 ijerph-17-06662-f002:**
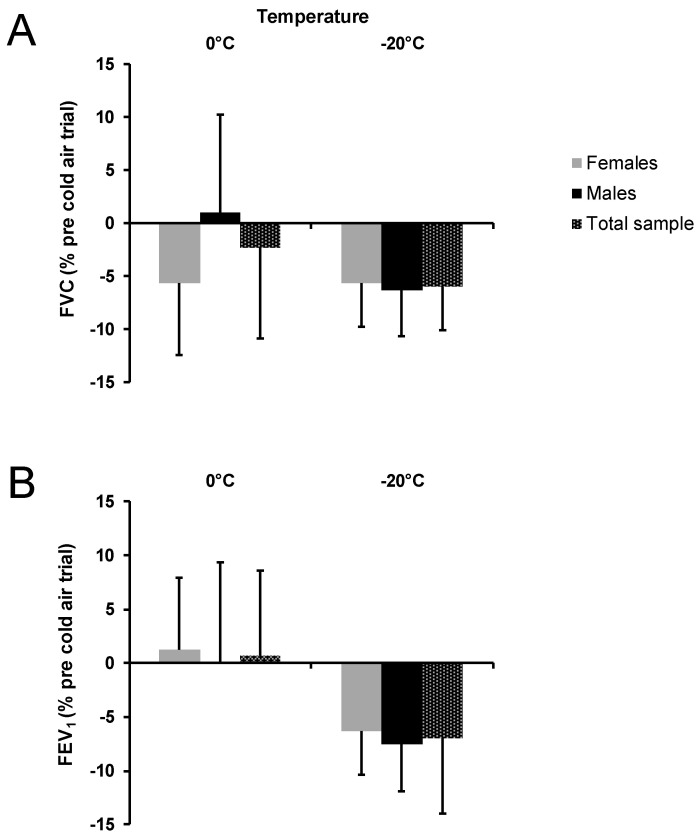
Acute respiratory function changes FVC (**A**) and FEV_1_ (**B**) expressed as % from pretrial values for the total sample and separately for females and males at 0 °C and at −20 °C. Negative values showed a decrease from pretrial values. FVC, forced vital capacity; FEV_1_, forced expiratory volume in 1 second; error bars indicate standard deviations.

**Figure 3 ijerph-17-06662-f003:**
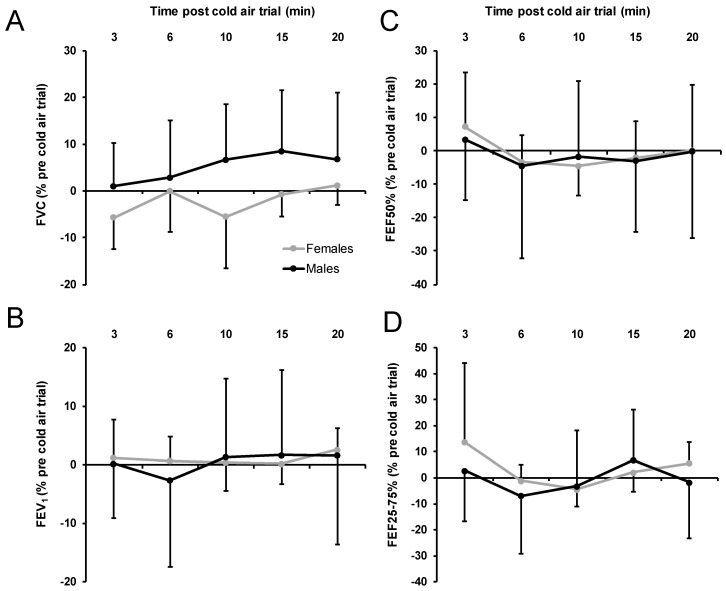
Acute respiratory recovery variables at 0 °C FVC (**A**), FEV_1_ (**B**), FEF50% (**C**), FEV25–75% (**D**) expressed as % from pretrial values separately for females and males. Negative values show a decrease from pretrial values. FVC, forced vital capacity; FEV_1_, forced expiratory volume in 1 second; FEF50%, forced expiratory flow at 50%, FEF25–75%, forced expiratory flow at 25–75%; error bars indicate standard deviations.

**Figure 4 ijerph-17-06662-f004:**
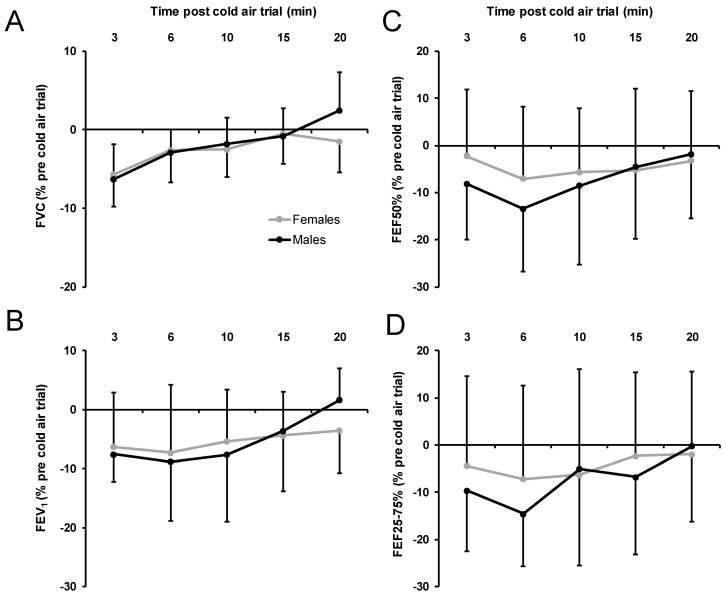
Acute respiratory recovery variables at −20 °C FVC (**A**), FEV_1_ (**B**), FEF50% (**C**), FEV25–75% (**D**) expressed as % from pretrial values separately for females and males. Negative values show a decrease from pretrial values. FVC, forced vital capacity; FEV_1_, forced expiratory volume in 1 second; FEF50%, forced expiratory flow at 50%, FEF25–75%, forced expiratory flow at 25–75%; error bars indicate standard deviations.

**Table 1 ijerph-17-06662-t001:** Descriptive anthropometric, graded exercise test and baseline respiratory function.

Variable	Female (*n* = 9)			Male (*n* = 9)			*p*-Value
	Mean	SD	(Range)	Mean	SD	(Range)	
Age (year)	24	3	(20–27)	24	2	(21–26)	0.83
Height (cm)	166	6	(158–172)	180	5	(177–184)	<0.01
Weight (kg)	59	6	(51–65)	75	5	(71–79)	<0.01
BMI (kg/m^2^ )	21	1	(20–23)	23	2	(22–25)	<0.01
Maximal oxygen uptake (mL/min/kg)	49	3	(45–56)	58	7	(50–70)	<0.01
Maximal heart rate (bpm)	196	7	(180–203)	200	7	(190–210)	0.19
Maximal rating of perceived exertion	19.2	0.4	(19–20)	19.1	0.3	(19–20)	0.73
Maximal speed (km/h)	11.0	1.0	(10–12)	13.5	1.0	(12–16)	<0.01
Baseline FEV_1_ (L)	4.1	1.0	(3.4–5.3)	6.0	0.5	(5.5–7.1)	<0.01
FEV_1_ (% predicted)	121.2	18.1	(93.5–151.2)	128.9	12.4	(115.0–148.4)	0.75
Baseline FVC (L)	5.3	1.1	(4.3–7.2)	7.2	0.8	(5.6–8.3)	<0.01
FVC (% predicted)	129.8	25.8	(86.5–169.3)	126.7	13.3	(100.3–142.2)	0.31
FEV_1_/FVC ratio	0.8	0.1	(0.6–0.9)	0.8	0.1	(0.7 – 1.0)	0.32
FEV_1_/FVC ratio (% predicted)	95.5	14.1	(68.5–113.0)	101.6	10.1	(92.2–125.6)	0.31
Baseline FEF50 (L/min)	4.7	1.0	(3.3–6.31)	7.1	1.0	(5.6–8.7)	<0.01
Baseline FEF25–75 (L/min)	4.0	1.1	(2.7–6.0)	6.2	1.2	(5.0–8.3)	<0.01
Baseline FEF25–75 (% predicted)	112.0	26.0	(78.7–158.7)	128.9	25.8	(103.0–173.7)	0.18

Data are reported as means, SD with ranges reported as minimum–maximum for female and male participants, where significant difference of *p* < 0.05 was considered different between sexes. Forced Expiratory Volume in the first second (FEV_1_), Forced Vital Capacity (FVC), Forced Expiratory Flow at 50% percent (FEF50%) and Forced Expiratory Flow between 25–75% (FEF25–75%) and FEV_1_/FVC ratios are shown in absolute units. Percentages predicted for age and sex are also shown for FEV_1_, FVC and FEV_1_/FVC ratio, as per guidelines [38].

**Table 2 ijerph-17-06662-t002:** Respiratory symptoms post-trial at 0 °C and −20 °C in female and male participants.

Temperature	Variable	Female (*n* = 9)	Male (*n* = 9)	*p*-Value	Effect Size
0 °C	Number of symptoms (M ± SD) (range)	1.2 ± 0.7 (0–2)	1.1 ± 1.2 (0–3)	0.796	0.10 ^a^
	Cough (%)	11	44	0.110	0.16 ^b^
	Wheeze (%)	0	0	<1.000	n.a. ^c^
	Chest tightness (%)	33	33	<1.000	1.00 ^b^
	Mucus (%)	78	33	0.058	7.00 ^b^
−20 °C	Number of symptoms (M ± SD) (range)	2.3 ± 1.0 (1–4)	2.4 ± 0.5 (2–3)	0.796	−0.13 ^a^
	Cough (%)	78	89	0.527	0.00 ^b^
	Wheeze (%)	11	0	0.303	n.a. ^c^
	Chest tightness (%)	78	78	<1.000	1.00 ^b^
	Mucus (%)	67	78	0.599	0.57 ^b^

Number of symptoms reported as mean ± SD and range (minimum and maximum) for females and males. Individual symptoms reported as relative frequencies for females and males. *p*-value for sex differences shown for each symptom and overall number of symptoms at each temperature. ^a^: Cohen’s d, ^b^: Odds ratio, ^c^: n.a., not applicable due to values of 0.

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
