# Peer review of "Are Respiratory Responses to Cold Air Exercise Different in Females Compared to Males? Implications for Exercise in Cold Air Environments"

_ijerph, 2020, doi:10.3390/ijerph17186662_

Round 1

Reviewer 1 Report

Dear Authors,

congrats for your work. 

Questions/recomendations:

INTRODUCTION

1.Why men and women should differ in respiratory responses during exercise in cold air enviroments?

2.Why its important to study this responses between men and women? Its just because we need to know if there are differences or if there are some specifications for each one?

You should answer this questions with more knowledge and more references

Methods/results/discussion

1.Did you control the typical type of exercise that your participants do?all your participants workout the same type of exercise? if no, does this not influenced your results?

2.Were they adapted to this type of environment? if yes, could this be one of the reason for your results between genders?

3. The lab conditions did not influence the results?

4. Why you have not collect body fat? the muscle mass could interfere in respiratory responses in cold air environment.

Author Response

Dear Authors,

congrats for your work.

Questions/recomendations:

INTRODUCTION

1.Why men and women should differ in respiratory responses during exercise in cold air environments?

At face value, we were not sure that there should be a difference in cold air exercise. This is a fundamental reason for why we decided to do the study (basically to answer a sex based environmental question that has not systematically been investigated yet). However, to extrapolate from the well-written work mostly from the W Sheel research group, which has found some sex-dependent mechanical lung differences in lab based exercise studies we reasoned that these differences may also extend to cold air. The differences reported by Dominelli et al. (2019) included smaller conducting airways in females compared to lung size-matched males. We hypothesized that these mechanical differences potentially exacerbate shear stress causing a greater responsiveness in cold air exercise in females compared to males. We feel like the lines 88 to 105 help provide the sex based reasons for how we arrived at our purpose and hope that it becomes clearer now how we have developed our reasoning for the study.

“Thus, the accumulated evidence to date does not provide clear evidence on how females differ in their acute response to cold air exercise compared to males. This is surprising given that the female lungs are more mechanically constrained inducing more shear stress in the airway at the same exercise intensity compared to males [19]. Some limited descriptive studies have shown that adult female athletes have significantly greater self-reported respiratory symptoms in a variety of HV sports compared to male athletes [20,21]; however sex based respiratory symptom differences have not been researched extensively either. Recent evidence also indicates pre-pubertal aged boy Nordic athletes (less than 13 years of age) have a greater self-reported wheeze than girls but this finding was related to greater training hours in boys compared to girls [22]. A recent field based study did find a more severe EIB response in females than males at -15°C [15] however, the majority of known sex based differences on respiratory function are based on warm weather or normal laboratory conditions [23,24]. Thus, there is enough preliminary evidence to hypothesize that sex based differences in EIB in cold environments may exist. Furthermore, this study also addresses the current paucity of sex based comparisons in exercise science [25] where this research can improve sex specific recommendations in sport science practice [26].”

2.Why its important to study this responses between men and women? Its just because we need to know if there are differences or if there are some specifications for each one?

You should answer this questions with more knowledge and more references.

As per our response above you are correct in the face value importance of understanding if there is a sex based difference in the respiratory response to cold air exercise. We could all agree that ventilating cold air at a high flow rate is a strong indirect provocation agent in humans. We also can see that in descriptive studies the frequency of reported respiratory symptoms is greater in female athletes (Langdeau et al., 2009) however in that study this was not specifically due to an acute exposure to cold air exercise. However, that study and the physiological studies around mechanical constraints in female lungs during exercise (see answer above), provide we feel sufficient reason to evaluate males and females in the same cold air environmental conditions. We have adapted the introduction slightly to address some of the descriptive level research, but are not sure what else to add because there really is no other research out there to add to what has already been stated including our own work in this area. We have added the descriptive research associated with sex based differences that we feel is relevant to this study in lines 81 – 86.

Methods/results/discussion

1.Did you control the typical type of exercise that your participants do?all your participants workout the same type of exercise? if no, does this not influenced your results?

We did not do an extensive background exercise history of individuals. However, they were all habitually active and were all used to hard exercise and sport effort to ensure they could tolerate the intense exercise protocol. We feel like we did a good job of excluding participants and of recruiting a relatively homogenous sample (i.e., maximal oxygen uptake (VO2max) > 50 ml/kg/min for males and > 45 ml/kg/min for females, no pre-existing acute or chronic diseases, no heightened symptoms of EIB and cough during low or moderate intensity exercise in cold weather, no previous history of adverse respiratory symptoms due to high intensity exercise, no regular smoking, no pregnancy, no lactation, please see lines 118 – 126). They all lived in the same geographic region thus their global exposure to the same environmental conditions was similar. We have not reported physical activity history in the results – but think this is a good addition. Please see the added sentence in the results (lines 245 – 249).

“The physical activity history and exercise history was similar across all participants and all participants were regularly doing hard exercise bouts / intervals as part of their university degree requirements or club based sports in the Tirol region of Austria. The participants all were living in the same geographic area and similar environmental exposure to outdoor ambient conditions.”

2.Were they adapted to this type of environment? if yes, could this be one of the reason for your results between genders?

They were all adapted to this environment – we have provided this as a statement in the results, please see comment before.  We agree this can be a potentially influencing factor, however there were no large differences in the type of female participant we recruited compared to male participant either in geographic region, upbringing or current exposure to long term outdoor severe temperatures. We pre-meditatively ensured we did not recruit participants who were outliers in extreme environmental conditions to ensure the generalizability of these results is to “healthy active adults”. We feel we have presented this adequately in the conclusion (l. 470 – 472).

“To elucidate, we feel that these results can be generalized to both females and males who live in winter or mountainous winter climates and are habitually physically active outdoors during cold weather months.”

  1. The lab conditions did not influence the results?

They did not – in fact that is a strength of the study – the chamber and lab environment (temperature and relative humidity) was precisely controlled and our results reflect our purpose of systematically evaluating sex based responses to cold air exercise.

  1. Why you have not collect body fat? the muscle mass could interfere in respiratory responses in cold air environment.

We did not collect body fat; however, the tight range of BMI (females: 20.3-22.6 kg/m², males: 21.6-25.2 kg/m²) indicates that muscle mass was similar in each sex (see Table 1). It is likely that males had more muscle mass overall as part of body composition differences between females and males. However, since we aimed to compare the acute respiratory responses (function, recovery and symptoms) between sexes, we did not consider body fat as a potential confounding variable in the sex comparison, but as part of the biological differences between sexes. We were more concerned about inter-individual differences in spirometry measures. Therefore, the outcomes for acute respiratory function and acute recovery were expressed as relative changes from pre-cold air trial value. This approach can be considered as a normalization procedure for inter-individual differences in spirometry measures.

We included the range of body height, body weight, and BMI to Table 1 to provide insight to the range of the sample.

Reviewer 2 Report

Manuscript Review – Are Respiratory responses to cold air exercise different in females….

Overall, this was a well done study and was well written. Thank you for your contributions to the field.

Much of the corrections are grammar fixes or providing clarity in presenting results (e.g., data in tables). I hope my comments below make sense and add to the clarity of your manuscript.

Introduction

  1. Line 36: insert the word “of” (“…via a number of mechanisms [3].”)
  2. Line 38-39: delete the last part of this sentence (“where a variety of contributing and predisposing factors related to EIB have been identified [3].”) as you already say something similar in line 36.
  3. Line 43: move “due to lack of standardized cold air conditions” to the end of the sentence.
  4. Line 65: grammar correction; “….because cold air is also dry where and drying…”
  5. Line 70: change “is” to “has”

Materials & Methods

  1. Line 168: delete “were” (“…time were participants were permitted…”)
  2. Line 172: do you have a reference for the validation you mention here?
  3. Line 193: grammar correction; “…in FEV1 and FVC were calculating calculated using the minimal post …”

Results

  1. Line 222: “…FEV1 and FVC but 4 female…”. Is this what you meant?
  2. Line 242: “…in FEV1 at -20°C compared to 0°” You already have -20°C at the beginning of the sentence.

Tables

  1. Table 1: The center justification makes it hard to read some of the data. You could create two columns under Female – one for the means and one for the standard deviations – and then do the same for the male data. Or space it out more or left justify so it is easier to read.
  2. Table 2: the first line for both 0°C and -20°C is crowded and the data run together. Space it out more for these two lines.

Discussion

  1. Line 375: You use a parenthesis before the word “(dry” but I don’t see the other “)”
  2. Line 378: put a comma after “This,”
  3. Line 394-396: Revise this sentence. I think I know what you are saying here but it reads awkwardly.
  4. Line 404: change “as” to “a”
  5. There is no specific limitations section. If the editor feels it is necessary it should be added to the manuscript.

References

  1. The following references have odd spacing: #2, 13, 18
  2. The following references have the first letter of all words in the article title capitalized: #23, 24, 25, 32, 50, 55
  3. Reference #32 = Authors names are capitalized

Author Response

Comments and Suggestions for Authors

Manuscript Review – Are Respiratory responses to cold air exercise different in females….

Overall, this was a well done study and was well written. Thank you for your contributions to the field.

Much of the corrections are grammar fixes or providing clarity in presenting results (e.g., data in tables). I hope my comments below make sense and add to the clarity of your manuscript.

We want to thank you for the careful reading of our manuscript. All your suggestions were helpful to make the manuscript much clearer and much easier to follow.

Introduction

Line 36: insert the word “of” (“…via a number of mechanisms [3].”)

Passage deleted in the revision process (based on the other Reviewers’ comments).

Line 38-39: delete the last part of this sentence (“where a variety of contributing and predisposing factors related to EIB have been identified [3].”) as you already say something similar in line 36.

Thank you; corrected as suggested.

Line 43: move “due to lack of standardized cold air conditions” to the end of the sentence.

Thank you; corrected as suggested.

Line 65: grammar correction; “….because cold air is also dry where and drying…”

We changed the sentence in the revision process (based on the other Reviewers’ comments):

“This study also found that female participants engaging in cold weather “High-Ventilation (HV)” sports had a greater EIB response, a common finding in HV athletes of both sexes compared to non HV athletes.

Line 70: change “is” to “has”

Sentence deleted in the revision process (based on the other Reviewers’ comments).

Materials & Methods

Line 168: delete “were” (“…time were participants were permitted…”)

Thank you; corrected as suggested.

Line 172: do you have a reference for the validation you mention here?

We have clarified what the validation is – we do not know of a published validation study on this device, but we regularly checked the accuracy of the spirometer using a 3 L calibration syringe throughout the study.

Line 193: grammar correction; “…in FEV1 and FVC were calculating calculated using the minimal post …”

Thank you; changed to: “…in FEV1 and FVC were calculated using the minimal post …”

Results

Line 222: “…FEV1 and FVC but 4 female…”. Is this what you meant?

Thank you; corrected as suggested.

Line 242: “…in FEV1 at -20°C compared to 0°” You already have -20°C at the beginning of the sentence.

Thank you; corrected as suggested.

Tables

Table 1: The center justification makes it hard to read some of the data. You could create two columns under Female – one for the means and one for the standard deviations – and then do the same for the male data. Or space it out more or left justify so it is easier to read.

Thank you; we followed your suggestion and created a column for the range, which we assess as important information in addition to the SD.

Variable

Female (n=9)

Male (n=9)

p-value

Mean

SD

(Range)

Mean

SD

(Range)

Age (year)

24

3

(20-27)

24

2

(21-26)

0.83

Height (cm)

166

6

(158-172)

180

5

(177-184)

<0.01

Weight (kg)

59

6

(51-65)

75

5

(71-79)

<0.01

BMI (kg/m²)

21

1

(20-23)

23

2

(22-25)

<0.01

Maximal oxygen uptake (ml/min/kg)

49

3

(45-56)

58

7

(50 – 70)

<0.01

Maximal heart rate (bpm)

196

7

(180-203)

200

7

(190-210)

0.19

Maximal rating of perceived exertion

19.2

0.4

(19-20)

19.1

0.3

(19-20)

0.73

Maximal speed (km/h)

11.0

1.0

(10-12)

13.5

1.0

(12-16)

<0.01

Baseline FEV1 (l)

4.1

1.0

(3.4-5.3)

6.0

0.5

(5.5-7.1)

<0.01

FEV1

(% predicted)

121.2

18.1

(93.5-151.2)

128.9

12.4

(115.0-148.4)

0.75

Baseline FVC (l)

5.3

1.1

(4.3-7.2)

7.2

0.8

(5.6-8.3)

<0.01

FVC

(% predicted)

129.8

25.8

(86.5-169.3)

126.7

13.3

(100.3-142.2)

0.31

FEV1/FVC ratio

0.8

0.1

(0.6-0.9)

0.8

0.1

(0.7 – 1.0)

0.32

FEV1/FVC ratio

(% predicted)

95.5

14.1

(68.5-113.0)

101.6

10.1

(92.2-125.6)

0.31

Baseline FEF50 (l/min)

4.7

1.0

(3.3-6.31)

7.1

1.0

(5.6-8.7)

<0.01

Baseline FEF25-75 (l/min)

4.0

1.1

(2.7-6.0)

6.2

1.2

(5.0-8.3)

<0.01

Baseline FEF25-75

(% predicted)

112.0

26.0

(78.7-158.7)

128.9

25.8

(103.0-173.7)

0.18

Table 2: the first line for both 0°C and -20°C is crowded and the data run together. Space it out more for these two lines.

Thank you; corrected as suggested.

Temperature

Variable

Female (n=9)

Male (n=9)

p-value

0°C

Number of symptoms (M ± SD)

(range)

1.2 ± 0.7

(0 – 2)

1.1 ± 1.2

(0 – 3)

0.796

Cough (%)

11

44

0.110

Wheeze (%)

0

0

< 1.000

Chest tightness (%)

33

33

< 1.000

Mucus (%)

78

33

0.058

-20°C

Number of symptoms (M ± SD)

(range)

2.3 ± 1.0

(1 – 4)

2.4 ± 0.5

(2 – 3)

0.796

Cough (%)

78

89

0.527

Wheeze (%)

11

0

0.303

Chest tightness (%)

78

78

< 1.000

Mucus (%)

67

78

0.599

Discussion

Line 375: You use a parenthesis before the word “(dry” but I don’t see the other “)”

The sentence now reads (l. 404-407):

“Generally speaking, an individual’s symptoms associated with intense cold air exercise varies largely and is substantially influenced by a number of factors such as athletic discipline where winter sport [48] or overall training hours per week [21] or the geographic location (dry inland winter environments have athletes with greater symptoms) [49].”

Line 378: put a comma after “This,”

Thank you; corrected as suggested.

Line 394-396: Revise this sentence. I think I know what you are saying here but it reads awkwardly.

We have looked at your comment and rewritten this statement – we agree that we could have been more clear and hope that the new sentences make our contrast of previous results to ours more clear (lines 425 – 431).

Line 404: change “as” to “a”

Thank you; corrected as suggested.

There is no specific limitations section. If the editor feels it is necessary it should be added to the manuscript.

We feel like we have woven in some limitations of the study into the discussion and hope that this addresses any concerns of not having a limitations section.

References

The following references have odd spacing: #2, 13, 18

Thank you for that careful observation. The odd spacing probably comes from the full justification format since only one blanc space is included in the references.

The following references have the first letter of all words in the article title capitalized: #23, 24, 25, 32, 50, 55

Reference #32 = Authors names are capitalized

Thank you; corrected as suggested.

Reviewer 3 Report

This is a well-written study examining respiratory responses following exercise at 0 and -20°.  The authors found no difference between women and men in the respiratory variables measured.

1.  The Introduction needs to specifically focus on women versus men differences.  Much of the section gives general background information on the respiratory responses to cold.  Cut back on this information and present the physiology that would potentially cause sex differences.  When the reader gets to the first sentence of the last paragraph in the Introduction, they don’t know how there may possibly be sex differences.  If there is a paucity of information on differences between women and men, that is okay.   The Introduction could then be cut by about a third. 

2.  The exercise routine to provoke EIB was essentially 8 minutes.  How generalizable is this to a real-world scenario where exercise time may be much longer and intensity just slightly lower than that used in this study? 

3.  Why is exercise below -20°C not recommended?  Are the responses that bad that warrant this recommendation? 

4.  On lines 243-245, switch the sentences for FVC and FEV1.  This is because the graph has FVC first and for the reader, if they look at the text and then the graph, they’ll be looking for the  FEV1 data to be first, but it is the second graph. 

5. On lines 371-372, add “at -20°C” to qualify this statement.  Could EIB occur at higher levels at lower temperatures? 

Author Response

Comments and Suggestions for Authors

This is a well-written study examining respiratory responses following exercise at 0 and -20°.  The authors found no difference between women and men in the respiratory variables measured.

  1. The Introduction needs to specifically focus on women versus men differences. Much of the section gives general background information on the respiratory responses to cold.  Cut back on this information and present the physiology that would potentially cause sex differences.  When the reader gets to the first sentence of the last paragraph in the Introduction, they don’t know how there may possibly be sex differences.  If there is a paucity of information on differences between women and men, that is okay.   The Introduction could then be cut by about a third. 

We took this comment to heart and have reduced the overall introduction about 1/3. Due to the recommendation of Reviewer #1, we have also added some sentences which strengthen the paucity of current sex based differences and highlighted the limited evidence that provided the purpose for our study. We think the introduction has been strengthened and we appreciate this insightful comment.

  1. The exercise routine to provoke EIB was essentially 8 minutes. How generalizable is this to a real-world scenario where exercise time may be much longer and intensity just slightly lower than that used in this study? 

We agree that the 8 min high intensity bout has limited generalizability to longer exercise bouts.  However,  we decided to go with the proven short high intensity provocation test to determine EIB that is often between 6 and 8 minutes long and has been used to induce heavy ventilation in a variety of conditions since the late 1970’s (often called the exercise provocation test). Thus, although we might have limited our generalizability to longer exercise bouts, we felt the exercise protocol we chose addressed the research aims we had by utilizing the proven and frequently used EIB provocation test.

  1. Why is exercise below -20°C not recommended? Are the responses that bad that warrant this recommendation? 

We considered our recommendation in the abstract and decided to alter the conclusion to better reflect the data. The present sentence is based on the results of our secondary analysis (e.g. significantly lower relative change in FEV1 at -20°C compared to 0°C in the total group) and fits better with our conclusions in the final paragraph of the discussion. We appreciate this comment and believe that the change to the final sentence of the abstract more appropriately reflects the conclusion of our results on the temperature comparison.

The sentence now reads as following (l. 26-28):

“However, intense exercise at -20°C is challenging to the respiratory system in both sexes and may lead to altered respiratory responses compared to mild winter conditions like 0°C.”

  1. On lines 243-245, switch the sentences for FVC and FEV1. This is because the graph has FVC first and for the reader, if they look at the text and then the graph, they’ll be looking for the  FEV1 data to be first, but it is the second graph. 

Thank you; corrected as suggested (l. 272-276).

“No significant temperature difference was found for FVC between -20°C (-6.0 ± 4.1%) compared to 0°C (-2.3 ± 8.5%), p = 0.327. The relative change in FEV1 was significantly lower at -20°C (-7.1 ± 1.7%) compared to 0°C (0.7 ± 7.8%) when males and females were combined, p = 0.003.”

  1. On lines 371-372, add “at -20°C” to qualify this statement. Could EIB occur at higher levels at lower temperatures?

Thank you; corrected as suggested.

Since we only studied the temperature of -20°C, we did not want to include information on the frequency of EIB at higher levels at lower temperatures to the manuscript.